# Identification of Retrocopies in Lepidoptera and Impact on Domestication of Silkworm

**DOI:** 10.3390/genes15121641

**Published:** 2024-12-21

**Authors:** Lingzi Bie, Jiahe Sun, Yi Wang, Chunfang Wang

**Affiliations:** 1Biological Science Research Center, Southwest University, Chongqing 400715, China; bielingzi@email.swu.edu.cn (L.B.); sunjiahe0502@email.swu.edu.cn (J.S.); 2Southwest University Hospital, Chongqing 400715, China

**Keywords:** lepidopteran insects, gene duplication, retrocopy, functional genomics, adaptive evolution

## Abstract

Background: During the domestication of silkworm, an economic insect, its physiological characteristics have changed greatly. RNA-based gene duplication, known as retrocopy, plays an important role in the formation of new genes and genome evolution, but the retrocopies of lepidopteran insects have not been fully identified and analyzed, which not only severely limits researchers from exploring the effects of retrocopies on lepidopteran insects but also affects the studies on the domestication of silkworm. Methods: We compared the genomes and proteomes of eight lepidopteran insects and used a series of screening criteria for auxiliary screening to obtain the retrocopies in lepidopteran insects and explored their characteristics. In addition, based on the silkworm transcriptome data from the SilkDB3.0 website, we explored the functions of the retrocopies on the domestication of the silkworm. Results: A total of 1993 retrocopies and 1208 parental genes in lepidopteran insects were obtained. We revealed that the retrocopies in Lepidoptera do not conform to the “out of X” hypothesis but fit the “out of testis” hypothesis. These retrocopies were subject to strong functional constraints and performed important functions in growth and development. Transcriptome analysis revealed that the expression pattern of the retrocopies and their parental genes were irrelevant. Through the analysis of the retrocopies in silkworm generated after domestication and located in the candidate domestication regions, the possible universal connection between the retrocopies and the domestication of silkworm were found. Conclusions: Our study pioneered the exploration of retrocopies in multiple Lepidoptera species and found the potential association between the retrocopies and the domestication of silkworm.

## 1. Introduction

Lepidoptera insects have a wide range of biodiversity and are the second largest insect order after Coleoptera. With recent advances in next-generation molecular sequencing technologies, research on Lepidoptera is quickly moving to a genomic scale. Lepidoptera genomes are relatively small in size and not as structurally complex as other eukaryotes. The genomes of some lepidopteran insects, such as silkworm, moth, and butterfly, have been published [1]. In lepidopteran insects, silkworm is very special, because it is the only domesticated species. The reproduction, morphology, behavior, and environment of silkworms have also undergone great changes [2]. Owing to artificial domestication, silkworm has a larger food intake and a smaller and safer living environment, so it has a larger silk yield, larger volume, degraded body color, and lower environmental sensitivity, compared with wild silkworm. In fact, the most fundamental reason is that the genetic material of the domesticated species has changed dramatically, and new genes have been formed during the long process of domestication and artificial selection [3].

Gene duplication is an important method of new gene formation and can be divided into RNA-mediated and DNA-mediated methods [4]. A retrocopy is a gene duplication formed by RNA-mediated retroposition, which refers to the reverse transcription of the mRNA formed by the parental gene to form cDNA under the action of the reverse transcriptase, after which the cDNA is inserted into a new position in the genome to form a new gene. Most retrocopies lose function due to improper insertion sites or sequence changes, which are called retropseudogenes. Some retrocopies form functionally active retrogenes because they retain the complete open reading frame (ORF) and obtain the normal promoter or enhancer sequences. Other genes bind to host gene sequences to form new coding regions known as chimeric retrogenes [4]. By recruiting new protein-coding regions, retrocopies were likely to evolve into new protein genes and drive genetic innovation and adaptive evolution in silkworm.

In recent studies, many retrocopies have been screened from the genomes of primates [5,6,7], fish [8,9], and mammals. The evolution of retrocopies in the genomes of higher animals has been preliminarily studied. In plants, retrocopies in the *Arabidopsis* [10], rice [11], *Populus* [12], and green algae [13] have been identified and analyzed. In insects, the identification of retrocopy is relatively rare, focusing only on individual model insects, such as *Drosophila* [14,15]. In Lepidoptera, only nocturnal moth [16] and silkworm [17,18] have identified retrocopies. In silkworm, two groups (Toups et al. and Jun Wang et al.) identified some retrocopies, but they did not explain the relationship between retrocopies and the domestication of silkworm. They mainly focus on whether there is excessive retroposition movement in the Z chromosome. Toups et al. [17] have identified 22 retrotransposition events and have found no excess retroposition movement out of the Z chromosome in the silkworm genome, while Jun Wang et al. [18] have identified 68 retrocopies and have found that excessive retrogenes move out of the Z chromosome. Some of the functions of retrocopies have been confirmed experimentally. For example, chondrodystrophy in dogs is caused by an overexpressed *Fgf4* retrocopy on chromosome 18 [19,20,21]. The *CG7804* retrocopy plays a role in multiple developmental stages (including the critical embryonic stage) in *Drosophila* [22]. *CYP98A8* and *CYP98A9* are involved in pollen development in *Arabidopsis* [23]. Therefore, the retrocopies are important driving forces for the origination of new genes and evidence for evolutionary innovation [5].

Therefore, we select the silkworm and other seven closely related lepidopteran insects and perform a systematic investigation of retrocopies in their genomes. We hope to identify the location and formation process of retrocopies in these eight lepidopteran insect genomes. Sequence diversity, the time of formation, and the orthologous and function of these retrocopies can be explored. We find the structural and functional characteristics of these retrocopies. Through the functional analysis of some special retrocopies in silkworm, the effect of retrocopies on the domestication of silkworm is found.

## 2. Materials and Methods

### 2.1. Data Sources

We selected eight species of Lepidoptera insects as the research subjects, including silkworm *Bombyx mori* (*B. mori*), wild silkworm *Bombyx mandarina* (*B. mandarina*), *Manduca sexta* (*M. sexta*), *Spodoptera litura* (*S. litura*), *Plutella xylostella* (*P. xylostella*), *Bicyclus anynana* (*B. anynana*), *Trichoplusia ni* (*T. ni*), and *Spodoptera frugiperda* (*S. frugiperda*).

The genome sequences, protein sequences, and transcriptome datasets of silkworm were downloaded from the SilkDB3.0 database (https://silkdb.bioinfotoolkits.net, accessed on 21 December 2024). The datasets of the other seven lepidopteran insects were downloaded from the NCBI database (https://www.ncbi.nlm.nih.gov/, accessed on 21 December 2024) (Appendix A).

### 2.2. Identifying Retrocopies in Lepidopteran Insect

Since the sequence of a retrocopy was a gene segment that had lost introns and was highly similar to its parental gene, its identification process in the genome generally relied on aligning the protein sequence with the genomic sequence to identify candidate retrocopies using LAST package (parameters: -p BL62- a 11- b 2- F 15- x 20- d 63- z 20) [24]. To avoid duplicate results, only the longest transcript of each gene was selected for comparison. According to the alignment results in the previous step, we set the criteria to screen the candidate retrocopies (identity of the sequence alignment was greater than 50% at the amino acid level, the coverage was greater than 50%, and minimum length was greater than 50 amino acids). When the distance between the candidate retrocopies was less than 40 bp, BEDTools (v2.29.2) [25] with default parameters was used to merge the adjacent candidate retrocopies. Next, LAST software (v1186) was used to align the merged sequence with the multiexon protein and retain the best hit as the candidate parental gene. Then, we estimated the number of missing introns on the basis of the comparison output file to obtain reliable results. We calculated the position of introns on the protein sequence on the basis of the annotation file and judged the number of introns missing from the parental gene of the best candidate. Finally, only the retrocopies which lost at least two introns and at least one intron more than 50 bp in length were retained. Some parental genes produced too many retrocopies, and there were false positives. If a parental gene produced more than 10 retrocopies, those parental genes and the retrocopies were deleted. We minimized the number of false positive retrocopies by comparing the retrocopies with the genome sequence, discarding the results that may involve DNA replication. If the retrocopies showed that there were multiple highly similar sequences in the genome (the identity was more than 80%, the coverage was more than 80%, and the sequence length of the protein was more than 80 amino acids), they were deleted. In addition, the identified retrocopies were classified based on their sequence characteristics. Retrocopies with premature stop codons or frameshift mutations were classified as retropseudogenes; otherwise, they were classified as intact retrocopies. If an intact retrocopy had multiple exons, it was classified as a chimeric retrogene; otherwise, it was classified as a single-exon retrocopy.

### 2.3. Sequence Features, Divergence Times, Potential Functions, and Orthology of Retrocopies Within Lepidopteran Insect Genomes

In the first step, in order to identify repeat elements from the insect genomes and ensure the accuracy of the results, RepeatModeler (v2.0.4) [26] with default parameters was used for the de novo modeling of the insect genomes. The model library was constructed taking the RepBase (v21.12) [27] known repeat library into consideration. In the second step, the BEDTools (v2.29.2) package was used to extract the retrocopies and their upstream and downstream 10,000 bp (the sequence of this interval can include most of the functional and regulatory elements of the retrocopies) according to the start and end position of the retrocopies. In the third step, the RepeatMasker package (v4.0.7) [28] was used to analyzed the sequences information of the retrocopies.

Here, we calculated the Ka (nonsynonymous substitution rate), Ks (synonymous substitution rate), and Ka/Ks values between each retrocopy and its parental gene in order to obtain the age and potential function of the retrocopies. First, the CDS sequences of the retrocopies and their parental genes were extracted. We subsequently used ClustalW2 [29] to perform multiple alignments of retrocopies and their parental gene sequences. Finally, we used KaKs_calculator_2.0 [30] to calculate the Ka, Ks, and Ka/Ks values.

We can understand the evolutionary relationship of lepidopteran insects by analyzing their orthologous retrocopies. Species-specific and orthologous retrocopies of lepidopteran insects were identified by comparing the similarity. First, the sequences of the retrocopies were extracted from the genome via the BEDTools package according to the start and end positions of the retrocopies. We then applied the BLAT package (v351), parameters: -mask = lower; -tileSize = 12; -minScore = 50; -Min identity = 0) to align retrocopy sequences pairwise between all species [31]. Finally, according to the alignment results, if the identity was greater than 80% at the amino acid level, the coverage was greater than 80%, and if the length was greater than 50 bp, the two retrocopies were orthologous and shared. In contrast, the rest of the retrocopies were classified as species specific.

### 2.4. Analysis of the Potential Function of Retrocopy in B. mori

Gene expression research was an important prerequisite and method for studying the function of annotated genes. To estimate the expression of a retrocopy and its parental genes, we used HISAT2 (v2.2.1) [32], SAMtools (v1.9) [33], StringTie (v2.1.4) [34], and Ballgown (v 2.10.0) [35] software for analysis on the basis of the position information and transcriptome data. Because some retrocopies were highly similar to parental genes, the process used to analyze their expression could ensure that the transcriptome sequence could be aligned to the correct position. First, HISAT2 mapped the sequence to the corresponding annotation sequence, and SAMtools converted the SAM file to the BAM file and sorted it by name. To discriminate the transcription between the parental gene and the retrocopy, we used SAM’s NH:i:1 flag to ensure the reads mapped to a unique location on the genome. Then, we used StringTie software to calculate the expression value of the retrocopy and its parental gene. Finally, we used the Ballgown software to analyze the expression data results. The expression values of the retrocopy and its parental gene were expressed in fragments per kilobase of transcript per million fragments mapped (FPKM).

Owing to the limited annotations of the silkworm genome, all retrocopies were first converted into fruit fly homologous genes through the Ensembl BioMart database. The Gene Ontology (GO), and the Kyoto Encyclopedia of Genes and Genomes (KEGG) pathway enrichment analyses were subsequently performed with UniProt (https://www.uniprot.org/, accessed on 21 December 2024) with the default parameters. The statistical analysis related to functional enrichment was conducted using the R package clusterProfiler in R (version 3.6.3) with default parameters, and clustering results with *p* < 0.05 and FDR < 0.05 were selected as the research subjects.

### 2.5. RT-PCR

Total RNA was extracted from *B. mori* tissue samples via the TRIpure Total RNA Rapid Extraction Kit (Chengdu Biofit Biotechnologies Co., Ltd., Chengdu, China). We treated total RNA with DNase I (Chengdu Biofit Biotechnologies Co., Ltd., Chengdu, China) to remove genomic DNA to avoid contamination. Single-stranded cDNA was synthesized via reverse transcription amplification via the Goldenstar RT6 cDNA Synthesis Kit Ver 2. The cDNA products obtained from reverse transcription were diluted appropriately and used as the qPCR template for qPCR amplification. We identified the differential regions between the two retrocopies of interest (Retrocopy59 and Retrocopy152) and their parental genes and designed fluorescent quantitative PCR primers using online websites to amplify fragments between 100 and 200 bp. The primer sequence is as follows: Retrocopy59-R (ACCAATTACGCGAATCACGG), Retrocopy59-F (ACAACATGGCAAGCACTCTG) and Retrocopy152-R (ATCTTCCGTCCGGACAACTT), and Retrocopy152-F (AATCCCTGGAGGCAATCACA). Using Rpl3F (CGGTTGTTGGATATTGAG) and Rpl3R (GCTCATCCTGCCATTCTACT) as internal references, each gene was subjected to three independent experimental replicates, and the relative expression levels of the genes were calculated using the 2^−ΔΔCt^ method.

## 3. Results

### 3.1. Identification of Lepidopteran Retrocopies

The origin of new genes based on RNA replication has been an important driver of biological evolution [36,37]. We developed a set of pipelines to identify retrocopies and their parental genes by comparing the genomes with the protein sequences of all eight studied species (for details, see Section 2). A total of 1993 retrocopies and 1208 parental genes were obtained (Figure 1A), indicating that one parental gene can produce an average of 1.65 retrocopies. The number of retrocopies in Lepidoptera ranged from 185 (*B. mandarina*) to 316 (*S. frugiperda*). This result was much lower than the number of retrocopies in human (3590) [7], dog (3025) [38], and other species but greater than *Drosophila* (24) [39].

Reverse transcriptase, which had endonuclease activity and can recognize polyA tails was one of the key enzymes in the formation of retrocopies and was derived from a variety of retrotransposable elements [40]. The retrotransposable elements were usually located upstream and downstream of the retrocopies, mainly SINEs, LINEs, and LTRs. In this study, retrocopies in the Lepidoptera genomes were identified by retrotransposable elements. The result revealed that the repeat sequences upstream and downstream of retrocopies in Lepidoptera included mainly SINEs, LINEs and LTRs, the proportions of which are shown in Figure 1B, indicating that the retrotransposition events in Lepidoptera were mediated mainly by non-LTR retrotransposable elements. In mammals and plants, retrotransposition events are similarly mediated mainly by LINEs [40,41]. Therefore, non-LTR was more common in retrotransposition events.

We subsequently classified these retrocopies. Usually, the retrocopies only retained the coding region and lacked the regulatory elements of their parental genes, which led to an increase in nonsense mutations and frameshift mutations, and eventually they became retropseudogenes. To obtain expression ability, retrocopies needed to recruit new regulatory elements to evolve into functional retrocopies, which were defined as intact retrogenes. Approximately 95.5% (1904/1993) of retrocopies did not exhibit frameshift mutations or premature termination codons and were considered as intact retrogenes; otherwise, the remaining 89 retrocopies were defined as retropseudogenes (Figure 1C). This ratio was quite high compared with dog (476/3025) [38], zebrafish (440/652) [42], the East Asian nematode (29/43) [43], etc. In 1904 intact retrocopies, 230 retrocopies had not yet acquired regulatory elements or recruited new protein-coding regions. Among the remaining intact retrocopies, if it had a single exon, it was defined as a retrogene; if it was chimeric, it was considered as a chimeric retrogene. We identified 541 retrogenes and 1133 chimeric retrogenes (Figure 1C). By recruiting new protein-coding regions, single-exon retrocopies and chimeric retrocopies were likely to evolve into new protein genes and drive genetic innovation and adaptive evolution [37]. The retrocopies of lepidopteran insects were mainly single-exon retrocopies and chimeric retrocopies, accounting for 87.9% of all intact retrocopies. This result indicated that retrocopies in Lepidoptera had the potential to form functional genes in structure and made a certain contribution to new genes in Lepidoptera genomes. In the retrocopy dataset, we detected 149 retrocopies that did not overlap with any of the annotated genes, suggesting that they may be newly annotated genes or nonfunctional fragments. However, 99 of them were intact retrocopies and maintained the conserved open-reading-frame sequence of their parental genes. This result showed that the newly annotated retrocopies can not only shape the genome of lepidopteran insects but also complement blank positions that are not annotated.

There was a famous “out of X” hypothesis in the identification of retrocopies in many species, such as mammals [44], fruit fly [39], and chicken [45]. It meant that there was a significant excess of retrocopies that originated from the X chromosome and retroposed to autosomes, while new genes retroposed from autosomes were scarce [39]. But the silkworm had no X chromosome, it belonged to the zz/zw sex determination system. This study analyzed the chromosomal location distribution of retrocopies in silkworm and revealed that there were few retrocopies originated from the Z chromosome (Table 1), which indicated that the distribution of the retrocopies in silkworm did not have a sex chromosome preference and did not conform to the “out of X” hypothesis. The same was true of the other seven species of lepidoptera. To sum up, it was found that there was no sex chromosome preference in the distribution of retrocopies in lepidopteran insects. This may be a difference between the xx/xy and zz/zw genetic systems.

### 3.2. Expression and Functional Analysis of Lepidopteran Retrocopies

In lepidopteran insects, many retrocopies had the potential to express and formed functional genes in structure, so we examined whether these 1993 retrocopies were functionally restricted. A structural analysis revealed that retropseudogenes had no function, and only the intact retrocopies had the potential to perform functions because they retained the ORFs and obtained the normal promoter or enhancer. In genetics, Ka/Ks represented the ratio between the nonsynonymous substitution rate (Ka) and the synonymous substitution rate (Ks). This ratio can be used to determine whether there is selective pressure on this protein-coding gene. In the Ka/Ks analysis, if Ka/Ks < 0.5, it meant that the retrocopy had the potential to function; if Ka/Ks < 1, it indicated that the retrocopy was selected for purification; and if Ka/Ks > 1, it was subject to positive selection. The results revealed that in lepidopteran intact retrocopies, the Ka/Ks of less than 0.5 was up to 92.5%, and in retropseudogenes it was 74.2% (Table 2). It indicated that the two types of retrocopies had the potential to perform functions, but the ratio of intact retrocopies was larger, and it meant that the intact retrocopies were subject to more functional constraints. The mean Ka/Ks of the retropseudogenes (0.9146) was greater than the intact retrocopies (0.4666), and their Ka/Ks ratios clearly differed in terms of distribution. Thus, these results suggested that most retrocopies in the Lepidoptera genome had a Ka/Ks ratio of less than 1, indicating that they were subject to purifying selection (Figure 1D). The Ka/Ks ratio of 91.7% retrocopies was less than 0.5, indicating that most retrocopies in Lepidoptera were strictly functionally restricted (Figure 1D).

We performed a functional analysis of the retrocopy dataset with Ka/Ks less than 0.5 and found that it did perform some functions. GO and KEGG pathway enrichment analyses were used to functionally classify intact retrocopies with Ka/Ks values of less than 0.5. The functions of these 1761 retrocopies were detected. Notably, the most significant GO clusters were mainly concentrated on the growth process, immune process, and metabolic process of biological processes, including the cell cycle, signal transduction, defense responses to other organisms, and cellular metabolic processes, the metabolic process of nitrogenous compounds, the metabolic process of NADP, etc. (Appendix A). From the results, the functions of the retrocopies of Lepidoptera include several basic functions, which may include catalytic and hydrolase activity in the processes of growth and development, metabolism, immunity, and so on.

The KEGG molecular pathways of retrocopies of Lepidoptera were enriched mainly in cellular processes and pathways related to some metabolic processes (*p* < 0.05, Appendix A). It was noteworthy that the retrocopies in lepidopteran insects involved in metabolic process-related pathways were enriched in carbohydrate metabolism, energy metabolism, amino acid metabolism, and lipid metabolism (Appendix A). These genes mainly included the RNA polymerase gene family, the NADH dehydrogenase gene family, the ATP gene family, and the glutamine synthetase genes family. In conclusion, the retrocopies in lepidopteran insects may be involved in related pathways such as growth and development, metabolism, and immunity, and perform functions mainly in the form of various enzymes, contributing to the basic growth of these organisms.

### 3.3. Age Distribution and Evolutionary Analysis of Lepidopteran Retrocopies

To estimate the time of the origin of the retrocopies, we plotted the distribution of Ks values for the retrocopy and parental gene (Figure 2A). The Ks value represented the base substitution rate in the evolution process, and the Ks value can be used to infer the time of retrocopy formation. If the Ks value of a retrocopy was less than two, it indicated that the retrocopy was relatively young; otherwise, the retrocopy was considered old [9,45]. The results revealed that the retrocopies of Lepidoptera experienced two bursts (two peaks), with the first burst producing significantly more retrocopies than the second. This suggests that the majority of Lepidoptera retrocopies were ancient in origin. Then, we made further exploration on the relationship between Ks and Ka/Ks. The result showed that with the increase in Ks, the Ka/Ks value of the retrocopies tended to decrease (Figure 2B), indicating that the ancient retrocopies had greater functional limitations.

The retrocopies of the lepidopteran insects experienced two bursts and were formed mainly by an ancient retrotransposition event, indicating that the retrocopy of lepidopteran insects had an impact on evolution. Further analysis revealed that there were seven (<1%) orthologous retrocopies in the eight Lepidoptera species, indicating that these seven retrocopies could be traced back to before the differentiation of the eight Lepidoptera species (Figure 2C). A total of 130 orthologous retrocopies were identified between *S. frugiperda* and *S. litura*, and 117 orthologous retrocopies were identified between silkworm (*B. mori*) and wild silkworm (*B. mandarina*), indicating that more closely related species have more orthologous retrocopies. In addition, we also analyzed the species-specific retrocopies and revealed that the *P. xylostella* had the greatest number of unique retrocopies (213); the least unique retrocopies were found in the *B. mandarina* (72) (Figure 2C). These retrocopies were formed independently after the species and ancestral species diverged, which was an important start for the further exploration of Lepidoptera evolution.

We identified seven orthologous retrocopies, all of which were annotated as functional genes in the Lepidoptera genomes. They all were identified as intact retrocopies with functional potential, five of which were classified as chimeric retrocopies, while two of which were defined as single-exon retrocopies. During evolution, these seven retrocopies were old (Ks > 2) and functionally restricted by strict selectivity (Ka/Ks < 0.5). GO and KEGG pathway analyses revealed that these retrocopies were enriched in functions related to protein enzymatic activity and catabolic processes. These findings indicated that retrocopies shared by Lepidoptera had evolved into potentially functional genes during evolution.

We also estimated the rate of retrocopy origination (the average number of retrocopies that originated within each time period) during the evolution of Lepidoptera. The results showed that the highest rate of retrocopy origination reached 1.691 retrocopies per million years and the average rate of retrocopy origination was 0.811 retrocopies per million years (Table 3), which was much lower than human (68 retrocopies per million year), indicating that retrocopy formation was slower in Lepidoptera compared with primates, possibly due to the presence of many repetitive LINEs in the primate genome [46]. The retrocopy average origination rate of Lepidoptera was slightly higher than *Drosophila* (0.51 retrocopies per million years), but the difference between them was small, indicating that the rate of retrocopy origination in Lepidoptera was on the same order of magnitude as that in *Drosophila* [14]. These results demonstrated that retrocopies can not only introduce enormous genetic variations across lineages but also create many potentially functional new locations in the lepidopteran genomes.

### 3.4. Transcriptome and Functional Analysis of Silkworm Retrocopies

The origin of new genes provided critical genetic novelties for biological diversity and contributed to the evolution of lineage or species-specific phenotypic traits. As a domestic species, silkworm was one of the largest groups in terms of phenotypic diversity [47]. Therefore, this study used the transcriptome data of silkworm to comprehensively analyze the expression of retrocopies to understand the changes in the internal genetic material of specific tissues in different periods or the differences in retrocopies expression between different tissues in different periods, so as to further understand the expression of silkworm throughout its lifetime.

We used RNA-Seq data to detect the temporal expression dynamics of the retrocopies and the parental genes of the silkworm. A total of 253 silkworm transcriptome samples (for details, see Appendix A) were collected including 10 stages (instar, molting, wandering, pupa, etc.) and 16 tissues (silk gland, fat body, head, blood, ovary, testis, etc.). The results showed that most tissues had strong expressions on day 3 of the fourth larval instar (L4D3), fourth instar molting (L4_molting), day 0 of the fourth larval instar (L5D0), day 3 of the fifth larval instar (L5D3), wandering, and pre-pupa, which were also the prosperous periods of silkworm growth and development. Particularly, the high expression of the retrocopies in the silk gland including ASG (anterior silk gland), MSG (middle silk gland), and PSG (posterior silk gland) may indicate that the retrocopies play a role in the formation of the silk gland. For example, the highest expression (143 retrocopies) was found in the ASG during pre-pupa (Figure 3A). In the previous functional analysis of retrocopies, GO clusters were concentrated on the metabolic process of nitrogenous compounds and NADP, and KEGG molecular pathways were enriched in amino acid and energy metabolic pathways. Moreover, silk was mainly composed of protein and required energy to form. So, we speculated that some retrocopies such as 143 retrocopies in the ASG might form functional protein sequences related to silk gland growth, silk production, and cocooning. In addition, the retrocopies were expressed in fat body and testis at all stages. In fat body, the highest level was 139 retrocopies at the L5D3 larval stage; and in the testis, the highest level was 122 retrocopies that were transcribed during the wandering stage (Figure 3A). These results indicated that many retrocopies in the silkworm genome may play important roles in growth, the metabolic process, and the development of silkworm. It may represent a potential molecular basis for evolution in silkworm.

There was an “out of testis” hypothesis in the study of retrocopies. It meant that most retrocopies initially evolved into functional roles in testis, then transcribed in appropriate chromatin environments, and later evolved into more extensive expression patterns. In previous studies, such as human and *Drosophila*, retrocopies were first shown to be expressed in testis tissue [14,48]. We found that Lepidoptera also conformed to this hypothesis. In this research, the tissue-specific expression analysis of retrocopies in silkworm revealed that there were specific expressions in the testis at all stages, and the expression of retrocopies in testis was higher than in other tissues (Figure 3B). This study found that silkworm had an excess of retrocopies with testicular tissue-specific expression (4.83 times the average of other tissues), suggesting that retrocopies in silkworm may have initially been found to have transcriptional capabilities in the testis and subsequently evolved a broad expression pattern. These results indicated that the expression pattern of silkworm also had a testis-specific bias, suggesting that the expression of retrocopies in silkworm was in line with the “out of testis” hypothesis, which was consistent with the conclusions of primates and *Drosophila*.

Transcriptome data provided abundant information for elucidating the expression correlation of the retrocopies and parental genes. The expression correlation coefficient (R) of the retrocopies and parental genes was calculated. We considered the expression of a retrocopy as significantly associated with its parental gene if the FDR-adjusted *p* value was less than 0.05. In 215 pairs of genes of silkworm, one pair whose retrocopies did not express was deleted (FPKM value was 0). Among the remaining 214 gene pairs, 65 pairs were positively correlated (5 pairs were significantly positively correlated (R > 0.8, *p* < 0.05), 60 pairs were weakly positively correlated (0.3 < R < 0.8, *p* < 0.05)); 6 pairs were weakly negatively correlated (−0.8 < R < −0.3, *p* < 0.05); the other 143 gene pairs were not correlated (|R| < 0.3, *p* < 0.05, Pearson correlation test).

To study the relationship between evolutionary time and R, Ks was used as the value of differentiation duration. The following formula was used for conversion: Y = log(1 + R)/(1 − R). A positive linear regression of the Ks and Y values for each retrocopy revealed no correlation between these parameters (Figure 3C; R2 < 0.01, *p* = 0.168). In conclusion, there was no correlation between the expression patterns of the retrocopies and their parental genes over time.

Day 3 of the fifth larval instar (L5D3) was the most vigorous period for the development of the silkworm, and it can be seen from the above studies that the retrocopies were highly expressed in most tissues of L5D3, so we took this stage as an example to analyze the expression of the young retrocopies, the old retrocopies, and the parental genes. The results revealed that there were significant differences among these three groups, and compared with the young retrocopies, the expression pattern between the old retrocopies and the parental genes was more similar (Figure 3D). This result was consistent with the conclusions of primates and mammals [45]. These findings showed that the retrocopies of silkworm had acquired expression ability and may have evolved potential functions during evolution, and the old retrocopies were more likely to acquire functions.

### 3.5. Impact of Retrocopies on Silkworm Domestication

The silkworm (*B. mori*) was domesticated from the wild silkworm (*B. mandarina*) approximately 5000 years ago. Compared with the wild silkworm, the silkworm had changed in cocoon sizes, individual body sizes, growth rates, and digestion rates. Here, we found that the retrocopies played a role in the domestication of silkworm. We aligned the retrocopies of silkworm with the genome of wild silkworm and found that 25 expressed retrocopies were not orthologous to those in wild silkworm (L5D3 tissues were used as examples) (Figure 4A), which meant that they were produced after the domestication of silkworm, so we speculated that they may have performed functions in the domestication of silkworm. Among them, Retrocopy152 was highly expressed in the ASG and MSG of silkworm, indicating that it may have the function related to the domestication of silkworm like silk production and cocooning. And Retrocopy59 was widely expressed in various tissues of L5D3 which may play an important role in the growth and development of silkworm. In this study, the expression levels of the two retrocopies were verified via qPCR experiments, which revealed that the expression levels of the two retrocopies were consistent with the expression data of transcriptome (Figure 4B).

In 2014, Q. Xia et al. carried out a study on the silkworm domestication [49], using 29 silkworm genomes and 11 wild silkworm genomes for pangenome and pairwise linkage disequilibrium analysis. They found some regions in the silkworm genome, which had significant signatures of selective sweep, and defined these regions as genomic regions of selective signals (GROSS) or candidate domestication regions, and the genes in the region were defined as candidate domestication genes. Although many studies on domestication and retrocopies existed, there was no research on retrocopies in candidate domestication regions. Therefore, we focused on the effects of the retrocopies located in the GROSS of the silkworm genome.

We overlapped retrocopies with these regions and identified 8 retrocopies that overlapped with the candidate domestication regions (Table 4), which we defined as candidate domestication retrocopies. One of these retrocopies was identified as a retropseudogene that did not express and function and will not be discussed later. Of the remaining seven retrocopies, two were identified as single-exon retrocopies (retrogenes), and four were identified as chimeric genes. These seven retrocopies related to domestication were subject to strict selective functional constraints (Ka/Ks < 0.5), indicating they had potential functions. Protein function prediction revealed that these genes were enriched in functions related to protein enzymatic activity. GO enrichment analysis indicated that they were enriched in oxidative phosphorylation, protein synthesis and metabolism, ATP biosynthesis, etc. And molecular functions were enriched in protein folding and hydrolysis. According to the expression analysis, these seven intact retrocopies can be expressed, and the expression levels of Retrocopy68, Retrocopy3, and Retrocopy64 were relatively high (L5D3 tissues were used as examples), especially Retrocopy68, which was highly expressed in the various tissues of silkworm (Figure 4C), indicating that Retrocopy68 may have a basic function in growth and development. These results suggested that the retrocopies contributed to the growth and development of silkworm and played fundamental roles during domestication.

## 4. Discussion

Retrocopies are one of the main sources of new genes and significantly contribute to the plasticity of genomes. However, current in-depth research on retrocopies mainly focuses on mammals, plants, and fruit flies. In order to fill the research gap of retrocopies in Lepidoptera, this study used eight lepidopteran insects for the identification and analysis of retrocopies. By comparing the genomes and proteomes of eight lepidopteran insects, and using a series of screening criteria for auxiliary screening, 1993 retrocopies were accurately and efficiently obtained in the eight lepidopteran insect genomes, and the retrocopy dataset of lepidopteran insects was expanded. To our knowledge, this study provides for the first time a comprehensive and quite accurate dataset of retrocopies in eight lepidopteran insects, providing a data foundation for the further study of the retrocopies in Lepidoptera.

In species such as mammals, fruit flies, and chickens, the retrocopies conform to the “out of X” hypothesis, which means that there is a significant excess of retrocopies that originated from the X chromosome. Does this hypothesis also apply to organisms with zz/zw genetic systems? Previously, two groups of scientists have identified and studied the retrocopies in silkworm, which belongs to the zz/zw sex determination system. However, they drew completely opposite conclusions. The results of the study of Jun Wang et al. was that in the silkworm, 68 retrocopies were identified and too many retrocopies had been removed from the Z chromosome, and the “out of Z” retrocopies tended to produce ovarian-biased expression [18]. However, Toups et al. identified 22 retrocopies and did not find the excessive gene movements of the Z chromosome in the silkworm genome, and the final result was that birds and lepidopteran insects did not conform to the “out of X” hypothesis [17]. In our study, 215 retrocopies were identified in the silkworm genome, and the number of retrocopies was greatly increased compared with previously research. To verify the reliability of the retrocopies identified in this study, we examined the data sources and identification methods of Toups et al. and Jun Wang et al. The genomic data in the Toups’ study comes from KIKObase 2.0, and we found that the protein sequences in KIKObase 2 data contain a large number of stop codons, which may lead to a smaller number of retrocopies. They calculated local alignment scores for all pairs of peptide sequences within a species using mpiBLAST (v1.5.0). BlastP hits with a bit score of <200 were removed, and the remaining genes were clustered using MCL (v10.201). Then, clusters without both an intronless retrocopy and an intron-containing parental gene were excluded. The remaining clusters with only two genes were tabulated as a retrocopy–parent gene pairing [17]. We speculate that some mutations or deletions may have occurred in the retrocopies, which may reduce their overall similarity with the parental genes, affect the MCL clustering results, and may also lead to a low number of retrocopies. Jun Wang et al. aligned all peptide sequences against all peptide sequences with FASTA34 and screened out gene pairs based on a consistency of 40% and a coverage of 50%. Secondly, we classify candidate genes based on whether they contain transposable elements and set criteria for screening after classification. Finally, if the candidate gene does not have transposable elements and the candidate gene is a single-exon gene with highly similar multiexon genes, the screening results will be obtained [18]. It was known that 43.6% of the silkworm genome was occupied by transposable elements [50], and the proportion was much higher than *Drosophila* [51]. The upstream and downstream of retrocopies contain many transposable elements, and this identification method discards a large number of correct retrocopies, resulting in a lower number of retrocopies than the actual number in the genome, and may lead to some sequence fragments with low similarity being identified as retrocopies. Then, we checked the retrocopies of Toups and Jun Wang and found that some were misidentified as cases of retrocopies mostly due to sequence similarity produced by recent TE insertions into unrelated genes. Thus, it was very likely that the regions derived from transposable elements rather than real duplicated gene regions contributed to the similarity of gene families selected in their study. In addition, we noticed that in some cases, the alignment between the retrocopies and parental genes revealed no signature of intron loss, the hallmark of a retrotransposition event. Therefore, their dataset, after filtering out TE contamination, confirmed no excess of retroposition movement out of the Z chromosome in silkworm. An analysis of retrocopies from several other lepidopteran insects showed that there were also no extra retrocopies moving from the sex chromosomes. Therefore, the formation of the retrocopies of Lepidoptera do not conform to the “out of X” hypothesis. Up to this point, it seemed that this was a significant difference between the xx/xy and zz/zw systems.

Silkworm is the only domesticated model insect in lepidopteran insects, and it is a beneficial insect with great economic benefits. This study conducted research on the retrocopies in candidate domestication regions. The results showed that eight retrocopies annotated as functional genes were located in candidate domestication regions of the silkworm genome. And seven retrocopies were predicted to have enzyme activity and performed potential functions. We can boldly speculate that these seven retrocopies may have contributed to the adaptive evolution of silkworm domestication. For example, the growth and digestion rate of silkworm have significantly increased in the process of evolution. The retrocopies were highly expressed in the Malpighian tubule, which was the main excretory organ of arthropods. It is speculated that the retrocopies have contributed to the growth and metabolism of silkworm and may be involved in the growth and development process or metabolite excretion process of silkworm or may be a key regulatory factor that promotes the digestion process of silkworm. After all, retrocopies may have brought convenience and provided valuable resources for the adaptive evolution of silkworm.

## 5. Conclusions

Currently, research on retrocopies is a burgeoning field; however, investigations into the retrocopies of Lepidoptera are relatively scarce. Our study pioneers the exploration of retrocopies in multiple Lepidoptera species and examines the potential association between the retrocopies and the domestication of silkworm. Through our analysis of eight lepidopteran insect genomes, we have identified 1993 retrocopies and characterized their features and functions. Our findings indicate that the majority of Lepidoptera retrocopies are primarily mediated by non-LTR retrotransposable elements. These retrocopies do not align with the “out of X” hypothesis but are consistent with the “out of testis” hypothesis. The Ka/Ks ratios for 91.7% of the retrocopies are below 0.5, suggesting that the majority of retrocopies in Lepidoptera are subject to stringent functional constraints. Further functional and transcriptomic analyses reveal that the retrocopies significantly contribute to the growth and developmental processes of Lepidoptera. Furthermore, we have concentrated on the retrocopies that emerged following the domestication of silkworm and are located in the candidate domestication regions, indicating that certain retrocopies are highly expressed in the silk gland, and their functions are enriched in protein and energy metabolism. This may reveal that some retrocopies, such as Retrocopy152, may have contributed to novel phenotypic traits, including the enlargement of the silk gland and the enhancement of silk yield, during the domestication of silkworm.

## Figures and Tables

**Figure 1 genes-15-01641-f001:**
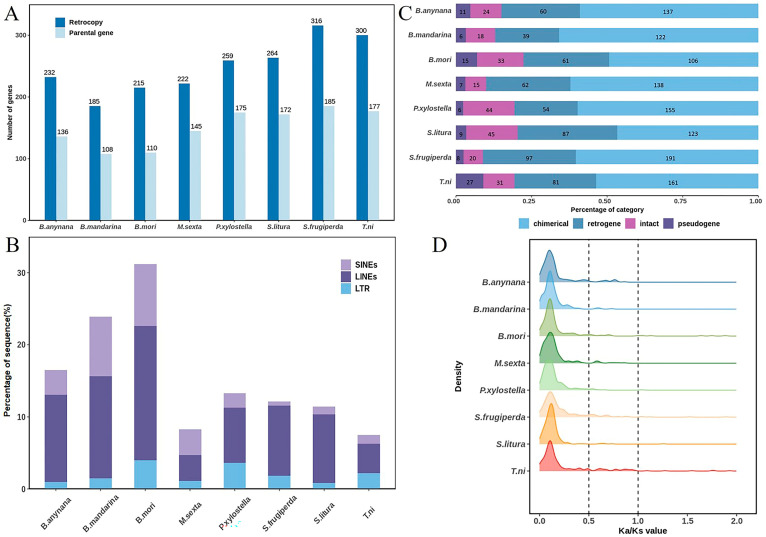
Retrocopies in the eight lepidopteran insect genomes. (**A**) Number of retrocopies and their parental genes. (**B**) Percentage of retrotransposable elements of retrocopies. (**C**) Percentage of types of retrocopies. (**D**) Distribution of ka/ks values of retrogenes and their parental genes.

**Figure 2 genes-15-01641-f002:**
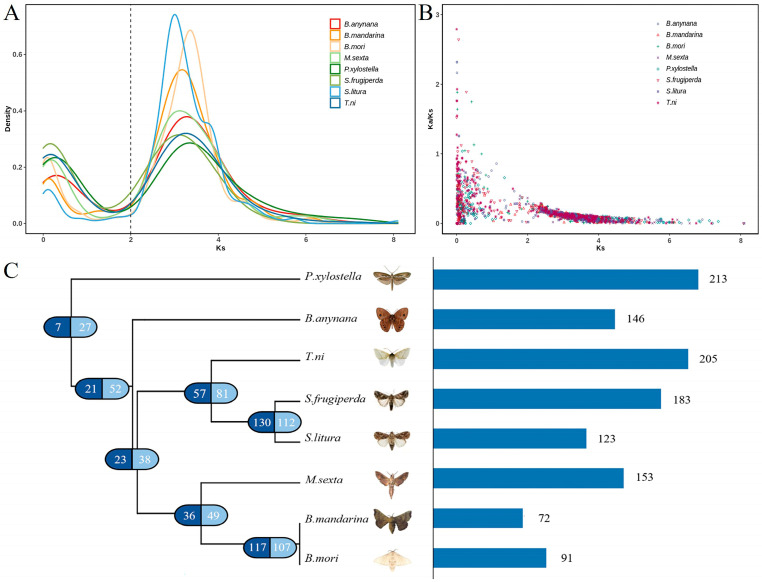
Retrocopy evolution within lepidopteran genomes. (**A**) Distribution of the Ks values. (**B**) Distribution relationship of the Ks values and Ka/Ks values. (**C**) Orthologous retrocopies and species-specific retrocopies. The left figure shows the number of orthologous retrocopies. The number in the dark blue semicircle on the branch represents the number of orthologous retrocopies behind the note on the tree, and the number in the light blue semicircle represents the number of orthologous retrocopies of one species behind the note and seven other lepidopteran species. The right figure shows the numbers of specific retrocopies in eight lepidopteran insects.

**Figure 3 genes-15-01641-f003:**
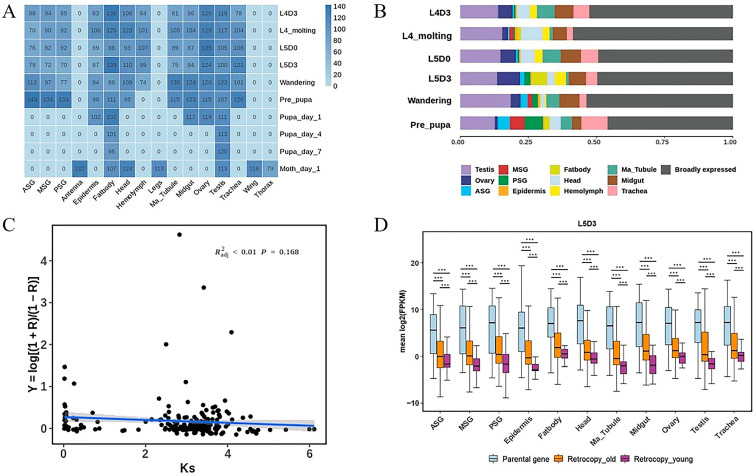
Expression analysis of retrocopies in silkworm. (**A**) The number of retrocopies of silkworm expressed in different tissues at different times. The numbers represent the level of retrocopy expression, and 0 meant that there are no transcriptome data in this tissue during this period. (**B**) Percentage of tissue-specific expression ability of retrocopies of silkworm in different tissues at different stages. (**C**) Correlation comparison of the expression patterns of the retrocopies and their parental genes in silkworm represented by Ks and Y = log(1 + R)/(1 − R). A positive linear regression of the Ks and Y values for each retrocopy revealed no correlation (the blue line). (**D**) Expression patterns of the retrocopies and the parental genes. *** indicates p < 0.001.

**Figure 4 genes-15-01641-f004:**
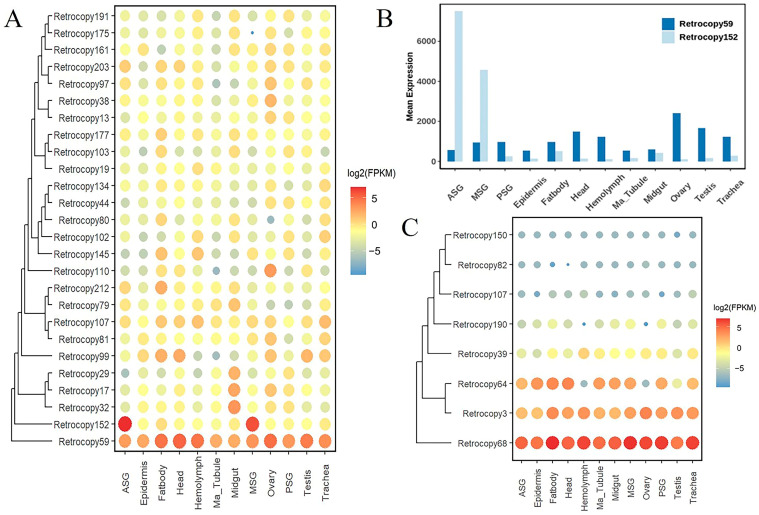
Relationship between retrocopies and silkworm domestication. (**A**) Expression values of 25 retrocopies. (**B**) Expression values of q-PCR of two retrocopies (Retrocopy52 and Retrocopy152). (**C**) Expression values of retrocopies in domestication candidate domestication regions of silkworm (L5D3).

**Table 1 genes-15-01641-t001:** Sources of genomes in Lepidopteran insect.

Movement	Expected	Observed	Excess (%)
Z→A	7	6	−14.28
A→Z	8	12	50
A→A	200	197	−1.5

**Table 2 genes-15-01641-t002:** Selective pressure of retrocopies in lepidopteran insect genomes.

Gene	Ka/Ks < 0.5	Ka/Ks = 0.5–1.0	Ka/Ks > 1.0	Mean Ka/Ks
Intact retrocopies	1761 (92.5%)	96 (5.04%)	33 (1.7%)	0.4666
Retropseudogenes	66 (74.2%)	18 (20.2%)	5 (5.6%)	0.9146

There are 13 intact retrocopies with Ka/Ks value of NA.

**Table 3 genes-15-01641-t003:** Estimated rate of retrocopy origination during lepidopteran evolution.

Evolutionary Period (Ma)	Branch Number	Number of Retrocopies	Evolutionary Period (Myr)	Average of Retrocopy (Myr)
114–156	8	14	42	0.333
111–114	7	2	3	0.667
67–111	6	13	44	0.295
60–111	5	34	51	0.667
16.9–60	4	73	43.1	1.691
0.0051–67	3	81	66.99	1.209

**Table 4 genes-15-01641-t004:** Retrocopies located in silkworm candidate domestication regions.

Retrocopy	Parental Gene	Host Gene	Gene Type	Functional
Retrocopy3	Chr15.332	Chr01.171	Retrogene	ubiquitin-protein transferase activity
Retrocopy39	Chr15.790	Chr05.600	Chimerical	peptidyl-prolyl cis-trans isomerase
Retrocopy64	Chr24.497	Chr09.305	Retrogene	protein binding
Retrocopy68	Chr19.416	Chr09.504	Chimerical	peptidyl-prolyl cis-trans isomerase
Retrocopy82	Chr01.34	Chr11.621	Chimerical	involved in glutamine metabolism
Retrocopy107	Chr07.550	Chr15.738	Intact retrocopy	uncharacterized protein
Retrocopy150	Chr22.674	NA	Retropseudogene	uncharacterized protein
Retrocopy190	Chr11.228	Chr25.579	Chimerical	oxidoreductase activity

## Data Availability

The original contributions presented in the study are included in the article/Appendix A, further inquiries can be directed to the corresponding authors.

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
