# Peer review of "Identification of Retrocopies in Lepidoptera and Impact on Domestication of Silkworm"

_genes, 2024, doi:10.3390/genes15121641_

Round 1

Reviewer 1 Report

Comments and Suggestions for Authors

Review

Title: Identification of Retrocopies in Lepidoptera and Impact on Do- 2 domestication of Silkworm

During the study, the genomes and proteomes of eight lepidopteran insects were compared and used a series of screening criteria for auxiliary screening to obtain the retrocopies in lepidopteran insects and explored their characteristics.

The authors made a new approach exploring the effects of retrocopies on lepidopteran insects.

The paper is well written, logic and adding new data to the scientific literature.

I appreciated the clear description pf the methods, and the results, and also the style of the figures, which makes the whole manuscript clear and understandable.

I have no specific comments on this paper, however I guess is the second time in my 25 year carrier when I can recommend paper in its present form.

With many best wishes

Author Response

Thank you very much for your positive review and valuable comments on our paper. We are delighted to hear that you found our approach novel and the paper well-written and logical.

We are especially glad that you appreciated the clarity of our methods, results, and figures. We put a lot of effort into presenting our research in a clear and understandable way, and it is gratifying to know that it has been effective.

Your recommendation of our paper in its present form is truly an honor and a source of motivation for us. We will continue to strive for excellence in our future research.

Once again, we thank you for your time and expertise in reviewing our work.

Reviewer 2 Report

Comments and Suggestions for Authors

Review Report

The manuscript investigates the role of retrocopies in the domestication of the silkworm, providing valuable insights into the genomic and evolutionary characteristics of these genetic elements in Lepidoptera. This study is both novel and relevant, addressing a significant gap in our understanding of retrocopies and their potential contributions to the domestication process. The findings—particularly the association of retrocopies with the "out of testis" hypothesis and their potential role in domestication—are intriguing and contribute meaningfully to the broader field of genome evolution.

However, there are several areas where the manuscript can be improved to enhance clarity, methodological rigor, and the completeness of the analysis.

Comments for the Authors

While the introduction offers a comprehensive background on the study's subject matter, it contains excessive general information that detracts from the main objectives:

- Lines 32–36: The extensive discussion on the diversity and ecological significance of Lepidoptera, while informative, is somewhat tangential to the specific focus on retrocopies and silkworm domestication.

- Lines 49–57: The detailed mechanism of retrocopy formation is overly descriptive and does not directly integrate with the study’s goals.

Recommendation: Streamline the introduction by focusing more directly on retrocopies and their relevance to silkworm domestication. This will help maintain a clear connection between the background information and the study's objectives.

Materials and Methods

The section provides an overview of the techniques and tools used but has several shortcomings that could impact reproducibility, clarity, and scientific rigor:

Expression Analysis Workflow (Lines 150–160):

The methodology lacks an explanation of how expression values (FPKM) were compared between retrocopies and parental genes. There is no discussion of how potential alignment ambiguities between highly similar retrocopies and parental genes were resolved.

Recommendation: Provide detailed information on how expression values were compared and analyzed. Explain the steps taken to distinguish between highly similar sequences during alignment to ensure accurate expression quantification.

RT-PCR Methodology (Lines 168–174):

Critical details are missing, such as the specific primers used for amplification and how they distinguish between retrocopies and parental genes. Replication details, including the number of biological and technical replicates, are not provided. Statistical analysis methods for interpreting relative expression are absent.

Recommendation: Include specific information about primer design and validation. Provide details on the number of replicates and the statistical methods used to analyze the RT-PCR data, ensuring that the experiments can be accurately replicated and evaluated.

Statistical Analysis Using R (Line 166):

While R is mentioned as the statistical analysis platform, no specific details about the statistical tests, corrections for multiple testing, or software parameters are provided.

Recommendation: Specify the statistical tests performed, any corrections for multiple comparisons (e.g., false discovery rate adjustments), and the R packages or versions used. This information is crucial for the reproducibility and validation of the results.

In conclusion, the manuscript presents significant findings that have the potential to advance our understanding of genome evolution and the domestication of the silkworm. However, to reach its full potential and be suitable for publication, the manuscript requires major revisions. Addressing the issues outlined above will greatly improve the clarity, rigor, and overall quality of the study.

Reviewer 3 Report

Comments and Suggestions for Authors

During the domestication of the silkworm, an economically important insect, its physiological characteristics underwent significant changes. RNA-based gene duplication, known as retrocopy, plays a key role in the formation of new genes and genome evolution, yet retrocopies in Lepidoptera have not been fully identified and analyzed, which limits research on their impact on these organisms and the domestication of silkworms. In this study, the genomes and proteomes of eight Lepidoptera species were compared, identifying 1,993 retrocopies, whose function may be crucial for the adaptation of silkworms in breeding conditions.

Below, I will present the points that the authors should consider and address before the publication of this paper.

Introduction:

1.      The introduction and methods are mixed in the text. It might be worth clearly separating them.

Materials and Methods

1.      The criteria for selecting retrocopy candidates (e.g., sequence identity >50%, absence of at least two introns) are detailed, but is the >50% threshold appropriate? Does it not eliminate potential retrocopies with lower similarity that could be biologically significant?

2.      The description of the use of LAST, BEDTools, and RepeatModeler is correct, but it is worth noting the lack of detailed parameter settings for these tools (e.g., specification of e-value or accuracy thresholds).

3.      The description of the RT-PCR protocol is good, but there is a lack of information regarding the accuracy and specificity of the primers used for amplifying retrocopies and parental genes. It would be helpful to include details on the primer design process and how they were tested.

4.      There is a lack of clear information regarding the number of biological replicates and the statistical analysis of the results. It would be beneficial to include these details.

Results

1.      Does the use of HISAT2, SAMtools, and StringTie provide sufficient accuracy in distinguishing the expression of retrocopies from parental genes? The authors should consider adding the results of validation of the results using an independent technique, such as wet-lab experiments (e.g., CRISPR, knockout).

2.      It would be advisable to extend the comparison to other insect groups (e.g., Coleoptera or Hymenoptera) in order to better place the results in a broader evolutionary context.

Discussion:

1.       The number of retrocopies in the silkworm (68 vs. 22 vs. 215) has been discussed in detail, but the reasons for these discrepancies could be explained more clearly.

What specific filtering criteria were used? How did the previous identification methods differ? How does the larger number of retrocopies influence data interpretation and conclusions?

2.       The issue of transposons is well presented, but more technical details are needed: Which specific types of transposons were the most frequent "confounders") Did other organisms (e.g., Drosophila) face similar challenges, or is this issue specific to Lepidoptera?

Aside from the points mentioned above, I have nothing further to add.

Round 2

Reviewer 2 Report

Comments and Suggestions for Authors

The authors have addressed all my questions. I have no further comments and recommend that the article be accepted for publication!